# Cholesterol activates the G-protein coupled receptor Smoothened to promote Hedgehog signaling

Giovanni Luchetti[1,2†], Ria Sircar[1,2†], Jennifer H Kong[1,2], Sigrid Nachtergaele[1,2], Andreas Sagner[3], Eamon FX Byrne[4,5], Douglas F Covey[6], Christian Siebold[4,5*], Rajat Rohatgi[1,2*]

[1]Department of Biochemistry, Stanford University School of Medicine, Stanford, United States; [2]Department of Medicine, Stanford University School of Medicine, Stanford, United States; [3]Mill Hill Laboratory, The Francis Crick Institute, London, United Kingdom; [4]Division of Structural Biology, University of Oxford, Oxford, United Kingdom; [5]Wellcome Trust Centre for Human Genetics, University of Oxford, Oxford, United Kingdom; [6]Department of Developmental Biology, Washington University School of Medicine, St. Louis, United States

**Abstract** Cholesterol is necessary for the function of many G-protein coupled receptors (GPCRs). We find that cholesterol is not just necessary but also sufficient to activate signaling by the Hedgehog (Hh) pathway, a prominent cell-cell communication system in development. Cholesterol influences Hh signaling by directly activating Smoothened (SMO), an orphan GPCR that transmits the Hh signal across the membrane in all animals. Unlike many GPCRs, which are regulated by cholesterol through their heptahelical transmembrane domains, SMO is activated by cholesterol through its extracellular cysteine-rich domain (CRD). Residues shown to mediate cholesterol binding to the CRD in a recent structural analysis also dictate SMO activation, both in response to cholesterol and to native Hh ligands. Our results show that cholesterol can initiate signaling from the cell surface by engaging the extracellular domain of a GPCR and suggest that SMO activity may be regulated by local changes in cholesterol abundance or accessibility.

*For correspondence: christian@strubi.ox.ac.uk (CS); rrohatgi@stanford.edu (RR)

†These authors contributed equally to this work

Competing interests: The authors declare that no competing interests exist.

## Introduction

Cholesterol, which makes up nearly half of the lipid molecules in the plasma membrane of animal cells, can influence many signal transduction events at the cell surface. It plays an important role in modulating the function of cell-surface receptors, including G-protein coupled receptors (GPCRs), the largest class of receptors that transduce signals across the plasma membrane, and antigen receptors on immune cells (*Burger et al., 2000*; *Pucadyil and Chattopadhyay, 2006*; *Swamy et al., 2016*). The structures of several GPCRs reveal cholesterol molecules tightly associated with the heptahelical transmembrane domain (7TMD) (*Cherezov et al., 2007*; *Ruprecht et al., 2004*; *Wu et al., 2014*). Cholesterol can influence GPCR stability, oligomerization and ligand affinity (*Fahrenholz et al., 1995*; *Gimpl et al., 1997*; *Gimpl and Fahrenholz, 2002*; *Prasanna et al., 2014*; *Pucadyil and Chattopadhyay, 2004*). Cholesterol also organizes membrane microdomains, or 'rafts,' containing proteins and lipids that function as platforms for the detection and propagation of extracellular signals (*Lingwood and Simons, 2010*). In all of these cases cholesterol plays a permissive role; however, it is not sufficient to trigger signaling on its own. Could cholesterol play a more instructive role— is it sufficient, not just necessary, to initiate signaling from the plasma membrane?

**eLife digest** Cells must communicate with each other to coordinate the development of most tissues and organs. Damage to these communication systems is often seen in degenerative disorders and in cancer. The Hedgehog signaling pathway is one of a handful of these critical systems. Reduced Hedgehog signals can lead to birth defects, while excessive Hedgehog signals can lead to skin and brain cancers. Cells transmit the Hedgehog signal by releasing a protein into their surroundings, where it can influence neighboring cells. Despite years of study, it is not understood how the Hedgehog signal is transmitted from the outside to the inside of a receiving cell.

Studies first done in flies and subsequently confirmed in humans have shown that a protein called Smoothened is needed to transmit the Hedgehog signal across the membrane of receiving cells. But it was not known how Smoothened carries out this critical signaling step to influence gene activation inside the cell and consequently to change cell behavior.

Now, Luchetti, Sircar et al. find that cholesterol, an important component of the cell membrane, directly binds to Smoothened and changes its shape so that it can activate Hedgehog signaling components inside cells. The experiments made use of mouse cells, and the discovery shows that cholesterol may play a previously underappreciated role in cell-to-cell communication.

This newly discovered role for cholesterol has implications for diseases, including a unique set of developmental disorders caused by abnormalities in pathways that produce cholesterol in human cells. Furthermore, this unexpected insight into Smoothened's activity may be clinically important, because Smoothened can cause cancer when mutated and is the target of anti-cancer drugs that are being used in the clinic. Following on from these findings, a major step will be to uncover if and how Hedgehog signals regulate cholesterol to allow Smoothened to transmit these signals across the cell membrane.

We find that cholesterol can indeed play an instructive signaling role in the Hedgehog (Hh) pathway, an iconic signaling system that plays roles in development, regeneration, and cancer. Multiple seemingly unrelated links have been described between cholesterol and Hh signaling (summarized in [*Eaton, 2008*; *Incardona and Eaton, 2000*]). While the best-defined role for cholesterol is in the biogenesis of Hh ligands (*Porter et al., 1996*), it also plays an independent role in the reception of Hh signals. Pharmacological or genetic depletion of cholesterol reduces cellular responses to Hh ligands, which has led to the view that cholesterol is *permissive* for Hh signaling (*Blassberg et al., 2016*; *Cooper et al., 1998*; *Cooper et al., 2003*; *Incardona et al., 1998*; *Incardona and Roelink, 2000*). Distinct from these previous observations, we find that an acute increase in plasma membrane cholesterol is *sufficient* to activate Hh signaling. Thus, cholesterol can initiate signals from the cell surface by acting as an activating ligand for a GPCR family protein.

## Results

### Cholesterol is sufficient to activate the Hedgehog signaling pathway

While testing the effect of a panel of sterol lipids on Hh signaling in cultured fibroblasts, we made the serendipitous observation that cholesterol could induce the transcription of Hh target genes. Since cholesterol is very poorly soluble in aqueous media, we delivered it to cultured cells as an inclusion complex (hereafter called MβCD:cholesterol) with the cyclic oligosaccharide Methyl-β–cyclodextrin (MβCD) (*Zidovetzki and Levitan, 2007*). Throughout this paper, we state the concentration of MβCD in the MβCD:cholesterol complexes, since this concentration is known exactly; for saturated complexes, the molar concentration of cholesterol is predicted to be ~8–10-fold lower than that of MβCD (*Christian et al., 1997*; *Klein et al., 1995*). MβCD:cholesterol complexes have been shown to be the most effective way to rapidly increase cholesterol in the plasma membrane, the subcellular location for most transmembrane signaling events (*Christian et al., 1997*).

MβCD:cholesterol activated Hh signaling in NIH/3T3 cells and Mouse Embryonic Fibroblasts (MEFs), cultured cell lines that have been extensively used for mechanistic studies of the Hh pathway

(*Figure 1*). MβCD:cholesterol treatment activated the transcription of *Gli1* (*Figure 1A and B*), a direct Hh target gene used as a measure of signal strength, and also reduced protein levels of the repressor form of the transcription factor GLI3, a consequence of signaling known to be independent of transcription (*Figure 1B*). MβCD:cholesterol induced a concentration-dependent, bell-shaped Hh signaling response (*Figure 1A*). Low doses of MβCD:cholesterol, which have only a minor effect on signaling, also increased the potency of the native ligand SHH, as seen by a leftward shift in the SHH dose-response curve (*Figure 1C*).

Cholesterol can influence multiple cellular processes at short and long timescales, so we compared the kinetics of MβCD:cholesterol-induced activation of *Gli1* to (1) the kinetics of MβCD:cholesterol-mediated delivery of cholesterol to cells and to (2) the kinetics of SHH-induced *Gli1* expression. Cholesterol loading of cells by MβCD:cholesterol was nearly complete by 2 hr, as determined by a standard enzymatic assay for free (unesterified) cholesterol (*Figure 1D*). The increase in cellular levels of free cholesterol was also confirmed by the transcriptional suppression of genes encoding enzymes in the pathway for cholesterol biosynthesis (*Figure 1—figure supplement 1A*). Importantly, there was a significant increase in the accessible or chemically active (*Radhakrishnan and McConnell, 2000*) pool of cholesterol in the plasma membrane, as shown by increased cell-surface labeling with a cholesterol-binding toxin (Perfringolysin O (PFO), *Figure 1—figure supplement 1B*) (*Das et al., 2013*). The initial activation of *Gli1* by MβCD:cholesterol coincided with the loading of cells with cholesterol, starting at 2 hr (*Figure 1D*). The kinetics of *Gli1* induction by MβCD:cholesterol paralleled those of *Gli1* induction by the native ligand SHH, despite the fact the absolute levels of signaling were higher in response to SHH. The rapid Hh signaling response to cholesterol, temporally correlated with the acute increase in cholesterol levels in the plasma membrane, is unlikely to be mediated by indirect or secondary transcriptional effects.

It was important to distinguish signaling effects caused by MβCD from those caused by cholesterol itself, especially because MβCD has been proposed to enhance Hh signaling by extracting an inhibitory sterol from cells (*Sever et al., 2016*). Following a previously-described protocol (*Christian et al., 1997*), we treated fibroblasts with a series of MβCD complexes in which the MβCD concentration was held constant at 1.25 mM while the cholesterol concentration was varied. Under these conditions, Hh signaling activity increased in proportion to the amount of cholesterol in the MβCD:cholesterol complexes (*Figure 2A*). Thus, cholesterol must be the active ingredient in these complexes that activates Hh signaling.

To define the structural features of cholesterol required to activate Hh signaling, we used MβCD to deliver a panel of natural and synthetic analogs (*Figure 2B*). This experimental approach was inspired by previous studies of the cholesterol sensor SREBP cleavage-activating protein (SCAP) (*Brown et al., 2002*). The Hh signaling activity of cholesterol was exquisitely stereoselective— neither its enantiomer (*ent*-cholesterol) nor an epimer with an inverted configuration only at the 3-hydroxy position (*epi*-cholesterol) could activate Hh target genes (*Figure 2C*). Enantioselectivity was consistent with cholesterol acting through a chiral binding pocket on a protein target, rather than by altering membrane properties (*Covey, 2009*). Hh signaling activity was also lost when either the number or the position of double bonds in the tetracyclic sterol nucleus were altered in 7-dehydrocholesterol (7-DHC) and lathosterol, two endogenous biosynthetic precursors of cholesterol. Interestingly, desmosterol, another immediate biosynthetic precursor of cholesterol that contains an additional double-bond in the iso-octyl chain, retained signaling activity. This structure-activity relationship points to the tetracyclic ring, conserved between cholesterol and desmosterol, as the critical structural element required for activity. We cannot exclude the possibility that desmosterol activated signaling because it was rapidly converted to cholesterol in cells. These strict structural requirements suggest a specific, protein-mediated effect of cholesterol on the Hh signaling pathway and further exclude the possibility that signaling activity is due to extraction of an inhibitor from cells by MβCD (present at the same concentration in all the sterol complexes tested in *Figure 2C*).

MβCD:sterol inclusion complexes have been suggested to potentiate Hh signaling by depleting an inhibitory molecule through an exchange reaction (*Sever et al., 2016*). This model cannot explain our results because the concentration (*Figure 2A*) and structure (*Figure 2C*) of the sterol in the inclusion complex, despite an unchanging MβCD concentration, can modulate Hh signaling activity.

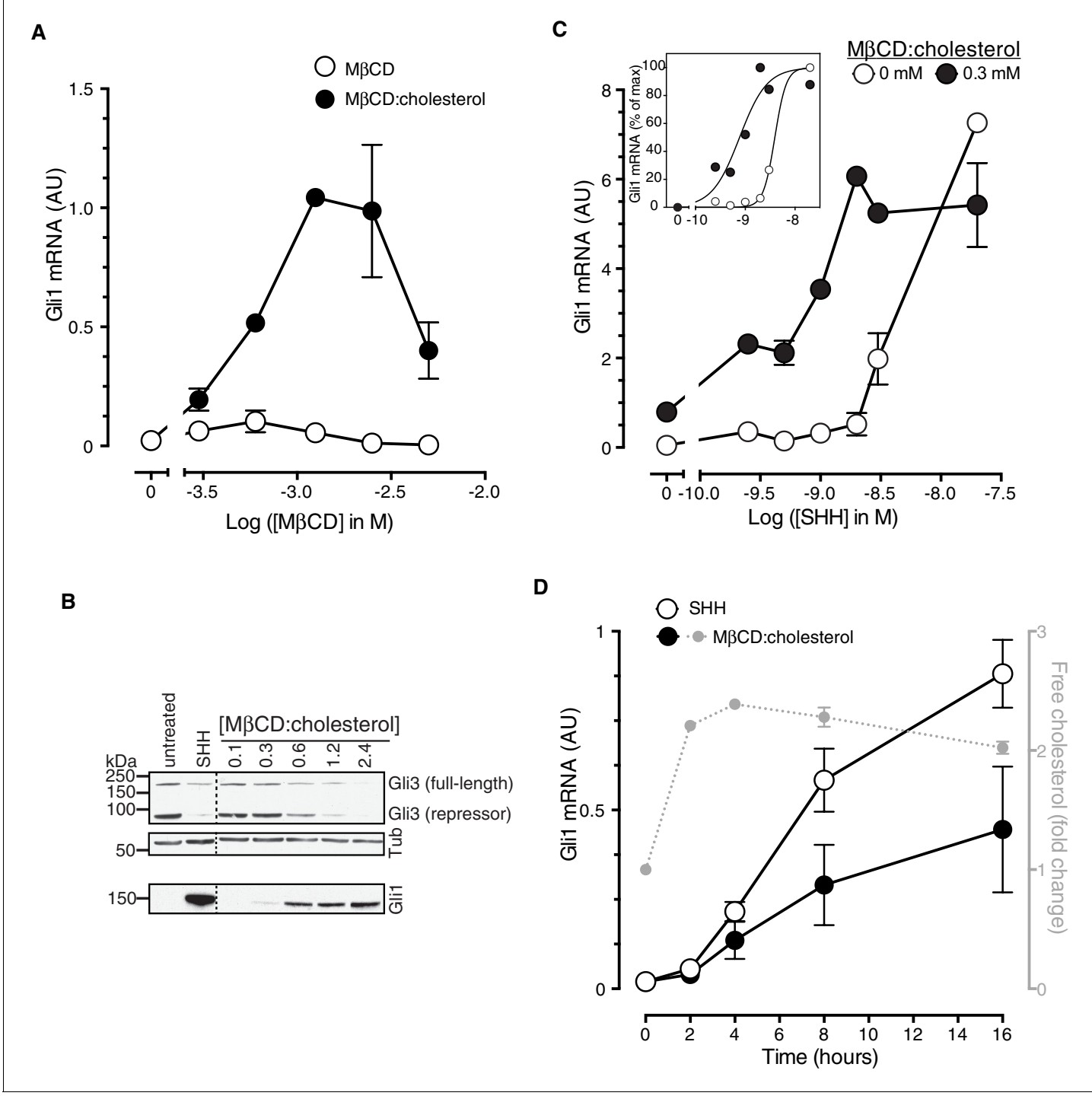

**Figure 1.** Cholesterol is sufficient to activate Hh target genes in NIH/3T3 cells. (**A**) *Gli1* mRNA, encoded by a direct Hh target gene, was measured by quantitative real-time reverse-transcription PCR (qRT-PCR) and normalized to mRNA levels of the housekeeping gene *GAPDH* after treatment (12 hr) with various doses of naked MβCD or a saturated MβCD:cholesterol (8.8:1 molar ratio) complex. In both cases, the concentration of MβCD is plotted on the abscissa. (**B**) Immunoblotting was used to measure protein levels of GLI1, full-length GLI3 and the repressor fragment of GLI3 after treatment (12 hr) with various concentrations (in mM) of MβCD:cholesterol. Dotted lines demarcate non-contiguous regions of the same immunoblot that were juxtaposed for clarity. (**C**) *Gli1* induction in response to various doses of SHH in the presence or absence of a low dose of MβCD:cholesterol. Inset shows non-linear curve fits to the data after a normalization in which the *Gli1* mRNA level in the absence of SHH was set to 0% and at the maximum dose of SHH was set to 100%. (**D**) Time course of *Gli1* induction (left y-axis) after treatment with SHH (265 nM) or the MβCD:cholesterol complex (2.5 mM). The gray circles (right y-axis) show the kinetics of increase in unesterified cholesterol (normalized to total protein) after the addition of MβCD:cholesterol. In all graphs, circles depict mean values from 3 replicates and error bars show the SD.

*Figure 1 continued on next page*

*Figure 1 continued*

The following figure supplement is available for figure 1:

**Figure supplement 1.** MβCD:cholesterol treatment increases the free cholesterol content of NIH/3T3 cells.

## Cholesterol functions at the level of Smoothened to activate Hedgehog signaling

A simplified schematic of the Hh signaling pathway is provided in *Figure 3A* (*Briscoe and Thérond, 2013*). The receptor for Hh ligands, Patched 1 (PTCH1), inhibits signaling by suppressing the activity of SMO, a member of the GPCR superfamily. SHH binds and inhibits PTCH1, thereby allowing SMO to adopt an active conformation and transmit the Hh signal across the plasma membrane. Cytoplasmic signals from SMO overcome two negative regulators of the pathway, protein kinase A (PKA) and suppressor of fused (SUFU), ultimately leading to the activation and nuclear translocation of the GLI family of Hh transcription factors.

To pinpoint the site of cholesterol action within this sequence of signaling events, we conducted a series of epistasis experiments (*Figure 3*). The addition of forskolin (Fsk), which leads to an increase in the activity of PKA, blocks Hh signaling at a step between SMO and the GLI proteins. Fsk inhibited MβCD:cholesterol-mediated signaling, placing the site of cholesterol action at the level of or upstream of PKA (*Figure 3B*). Two direct SMO antagonists, the steroidal natural product cyclopamine and the anti-cancer drug vismodegib, blocked *Gli1* activation by MβCD:cholesterol (*Figure 3B*) (*Sharpe et al., 2015*). This pharmacological profile established that MβCD:cholesterol requires SMO activity to promote signaling. Indeed, MEFs completely lacking SMO (*Smo*$^{-/-}$ cells) failed to respond to MβCD:cholesterol, and the stable re-expression of wild-type (WT) SMO, but not a point mutant locked in an inactive conformation (Smo-V333F), rescued signaling (*Figure 3C*) (*Varjosalo et al., 2006*; *Wang et al., 2014*). Thus, cholesterol must activate the Hh pathway at the level of PTCH1, SMO or an intermediate step.

We evaluated the possibility that MβCD:cholesterol interferes with the function of PTCH1 by using *Ptch1*$^{-/-}$ MEFs, which completely lack PTCH1 protein and have high levels of Hh target gene induction driven by constitutively activated SMO (*Taipale et al., 2002*). MβCD:cholesterol activated signaling in *Ptch1*$^{-/-}$ cells treated with cyclopamine to partially suppress SMO activity, showing that cholesterol signaling activity did not depend on the presence of PTCH1 (*Figure 3D*). MβCD:cholesterol behaved much like the direct SMO agonist SAG, since both could overcome SMO inhibition by cyclopamine in the absence of PTCH1.

Our epistasis experiments pointed to SMO as the target of cholesterol. However, compared to treatment with the native ligand SHH, SMO did not accumulate to high levels in primary cilia in cells treated with MβCD:cholesterol (*Figure 3—figure supplement 1A–C*), an observation that may explain the lower signaling efficacy of cholesterol compared to SHH.

## The cysteine-rich domain of Smoothened is required for the signaling activity of cholesterol

SMO contains two physically separable binding sites capable of interacting with steroidal ligands (*Figure 4A*) (*Nachtergaele et al., 2012*; *Sharpe et al., 2015*). Agonistic oxysterols, such as 20(S)-hydroxycholesterol (20(S)-OHC), engage a hydrophobic groove on the surface of the extracellular cysteine-rich domain (CRD) of SMO (*Myers et al., 2013*; *Nachtergaele et al., 2013*; *Nedelcu et al., 2013*). We recently reported that cholesterol could also occupy this CRD groove (*Byrne et al., 2016*). A cholesterol molecule was resolved in this groove in a crystal structure of SMO. Furthermore, purified SMO bound to beads covalently coupled to cholesterol and this interaction could be blocked by free 20(S)-OHC, consistent with the view that both 20(S)-OHC and cholesterol occupy the same binding site (*Byrne et al., 2016*). In addition, the extracellular end of the SMO 7TMD binds to the steroidal alkaloid cyclopamine, as well as to several non-steroidal synthetic agonists and antagonists (*Chen et al., 2002a*; *Chen et al., 2002b*; *Frank-Kamenetsky et al., 2002*; *Khaliullina et al., 2015*).

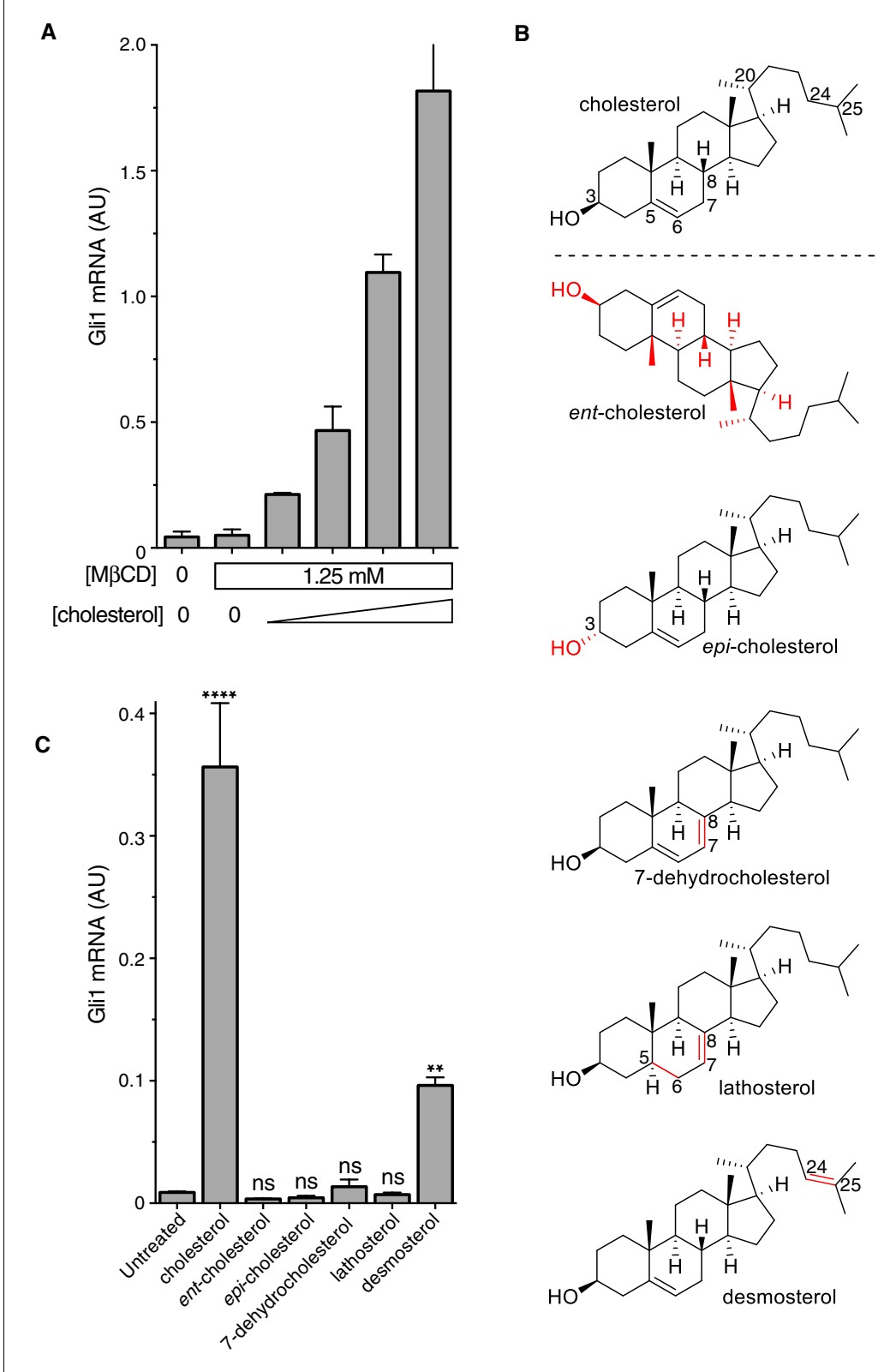

**Figure 2.** The cholesterol in MβCD:cholesterol complexes activates Hedgehog signaling. (**A**) Mean (±SD, n = 3) *Gli1* mRNA levels after 12 hr of treatment of NIH/3T3 cells with a series of inclusion complexes in which the MβCD concentration was clamped at 1.25 mM while the cholesterol concentration was varied to yield MβCD:cholesterol molar ratios of 12:1, 9:1, 7:1 and 6:1. (**B**) Structures of cholesterol analogs tested for Hh signaling activity as inclusion complexes with MβCD. Structural differences from cholesterol are highlighted in red: *ent*-cholesterol is the mirror-image of

*Figure 2 continued on next page*

Figure 2 continued

cholesterol with inverted stereochemistry at all 8 stereocenters; *epi*-cholesterol is a diastereomer with inverted stereochemistry only at the 3 carbon postion; 7-dehydrocholesterol, lathosterol and desmosterol are naturally occurring cholesterol precursors. (C) Mean (±SD, n = 4) *Gli1* mRNA levels after treatment (12 hr) with inclusion complexes of MβCD (1.25 mM) with the indicated sterols (see B for structures). Asterisks denote statistical significance for difference from the untreated sample using one-way ANOVA with a Holm-Sidak post-test.

In order to distinguish if the activating effect of cholesterol is mediated by the cholesterol binding groove in the SMO CRD or the cyclopamine binding site in the 7TMD, we asked whether MβCD:cholesterol could activate signaling in $Smo^{-/-}$ cells stably reconstituted with wild-type SMO (SMO-WT) or SMO variants carrying mutations in gatekeeper residues that have been shown to disrupt these two ligand-binding sites. The Asp477Gly mutation in the 7TM binding-site of SMO (*Figure 4A*), initially isolated from a patient whose tumor had become resistant to vismodegib, reduces binding and responsiveness to a subset of 7TM ligands, including SAG and vismodegib (*Yauch et al., 2009*). In the CRD, Asp99Ala/Tyr134Phe and Gly115Phe are mutations at opposite ends of the shallow sterol-binding groove that block the ability of 20(S)-OHC to both bind SMO and activate Hh signaling (*Figure 4A*) (*Nachtergaele et al., 2013*). The Asp99Ala and Tyr134Phe mutations disrupt a hydrogen-bonding network with the 3β-hydroxyl group of sterols (*Figure 4A*, inset) (*Byrne et al., 2016*).

The Asp477Gly mutation in the 7TMD domain had no effect on the ability of MβCD:cholesterol to activate Hh signaling (*Figure 4B*). SMO bearing a bulkier, charge-reversed mutation at this site (Asp477Arg) that increases constitutive signaling activity also remained responsive to MβCD:cholesterol (*Figure 4—figure supplement 1A*) (*Dijkgraaf et al., 2011*). In contrast, the Asp99Ala/Tyr134-Phe mutation in the CRD reduced the ability of MβCD:cholesterol to activate Hh signaling (*Figure 4C*). The Asp99Ala/Tyr134Phe SMO mutant was also impaired in its responsiveness to SHH and to 20(S)-OHC, but remained responsive to the 7TMD ligand SAG (*Figure 4C*). A complete deletion of the CRD (SMO-△CRD), which increased basal SMO signaling activity like the Asp477Arg mutation, also abolished signaling responses to MβCD:cholesterol (*Figure 4—figure supplement 1B*) (*Myers et al., 2013*; *Nedelcu et al., 2013*). This mutational analysis supports the model that the CRD binding-site, rather than the 7TMD binding-site, mediates the effect of cholesterol on SMO activity and thus on Hh signaling.

Interestingly, a mutation in Gly115, which is located on the opposite end of the CRD ligand-binding groove (*Figure 4A*), did not alter the response to MβCD:cholesterol, even though it diminished the response to 20(S)-OHC as previously noted (*Figure 4D*) (*Nachtergaele et al., 2013*). The SMO-Gly115Phe mutant also responded normally to the native ligand SHH (*Figure 4D*). Gly115 is located near the iso-octyl chain of cholesterol in the SMO structure (*Figure 4A*). The introduction of a bulky, hydrophobic phenyl group at residue 115 may prevent the hydroxyl in the iso-octyl chain of 20(S)-OHC from being accommodated in the binding groove, but not disrupt binding of the purely hydrophobic iso-octyl chain of cholesterol. The ability of mutations to segregate 20(S)-OHC responses from cholesterol responses is consistent with solution-state small-angle X-Ray scattering data showing distinct conformations for SMO bound to these two steroidal ligands (*Byrne et al., 2016*).

The ability of the Gly115Phe mutation to distinguish between cholesterol and 20(S)-OHC responses allowed us to address an important outstanding question: could cholesterol activate SMO only after being oxidized to a side-chain oxysterol? In addition to 20(S)-OHC, oxysterols carrying hydroxyl groups on the 25 and 27 positions can bind and activate SMO (*Corcoran and Scott, 2006*; *Dwyer et al., 2007*; *Myers et al., 2013*; *Nachtergaele et al., 2012*). However, 20(S)-OHC, 25-OHC and 27-OHC, when delivered to cells as MβCD conjugates, were all significantly compromised in their ability to activate Hh signaling in cells expressing SMO-Gly115Phe (*Figure 4—figure supplement 1C*). In contrast, cholesterol-induced signaling was unaffected (*Figure 4—figure supplement 1D*); therefore, cholesterol must not be activating signaling by being metabolized to one of these side-chain oxysterols. Instead, our data suggests that cholesterol can directly activate Hh signaling through the CRD of SMO.

## Cholesterol can drive the differentiation of spinal cord progenitors

Our mechanistic experiments in cultured fibroblasts led us to ask whether cholesterol could also promote Hh-dependent cell differentiation decisions. In the developing vertebrate spinal cord, the Hh

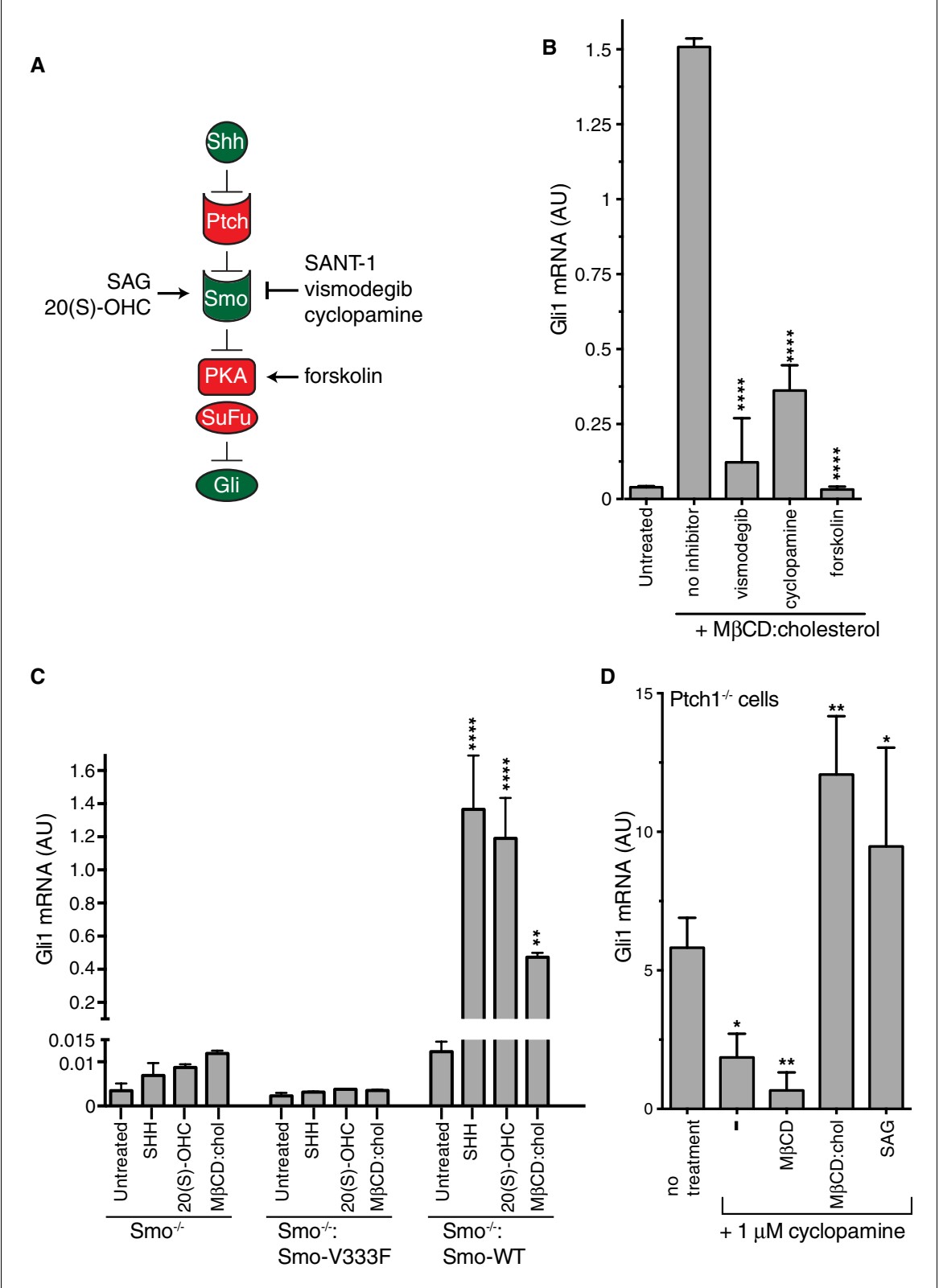

**Figure 3.** Smoothened activity is necessary for cholesterol to activate Hh signaling. (**A**) Schematic of the Hh signaling pathway showing the sequence in which core components function to transmit the signal from the cell surface to the nucleus. SAG and 20(S)-OHC are agonists and SANT-1, vismodegib, and cyclopamine are antagonists that bind and modulate the activity of SMO. Forskolin blocks signaling by elevating cAMP levels, which increases the activity of Protein Kinase A. (**B**) Mean (±SD, n = 3) *Gli1* mRNA levels after treatment with MβCD:cholesterol (1.25 mM, 12 hr) in the presence of
*Figure 3 continued on next page*

*Figure 3 continued*

vismodegib (1 µM), cyclopamine (10 µM) or forskolin (10 µM). (C) Mean (±SD, n = 3) *Gli1* mRNA levels after addition of agonists (12 hr) to Smo⁻/⁻ cells, in which both *Smo* alleles have been genetically inactivated, or Smo⁻/⁻ cells stably expressing a wild-type (WT) SMO protein or a variant SMO protein carrying an inactivating mutation (V333F) in its 7TMD (*Byrne et al., 2016*). SHH was used at 265 nM, 20(S)-OHC at 5 µM, and MβCD:cholesterol at 1.25 mM. (D) Mean (±SD, n = 4) *Gli1* mRNA levels in Ptch1⁻/⁻ cells after treatment with cyclopamine alone or cyclopamine in the presence of SAG (100 nM), MβCD (1.25 mM) or MβCD:cholesterol (1.25 mM). Asterisks denote statistical significance for differences from the "no inhibitor" sample in B, the V333F-expressing cell line in C, and the "no treatment" sample in D using one-way (B, D) or two-way (C) ANOVA with a Holm-Sidak post-test.

The following figure supplement is available for figure 3:

**Figure supplement 1.** MβCD:cholesterol fails to drive SMO accumulation in the ciliary membrane.

ligand Sonic Hedgehog (SHH) acts as a morphogen to specify the dorsal-ventral pattern of progenitor subtypes (*Figure 5A*)(*Jessell, 2000*). This spatial patterning process can be recapitulated *in vitro*. Mouse neural progenitors exposed to increasing concentrations of SHH will express transcription factors that mark differentiation towards progressively more ventral neural subtypes: low, medium and high Hh signaling will generate progenitor subtypes positive for Nkx6.1, Olig2, and Nkx2.2, respectively (*Dessaud et al., 2008*; *Gouti et al., 2014*; *Kutejova et al., 2016*).

MβCD:cholesterol induced the formation of both Nkx6.1⁺ and Olig2⁺ progenitor subtypes at a low frequency in cultures of mouse spinal cord progenitors (*Figure 5B and C*) and also activated the transcription of *Gli1* (*Figure 5D*). The activation of both *Gli1* induction and ventral neural specification by MβCD:cholesterol was significantly less than that produced by a saturating concentration of SHH. However, we note that MβCD:cholesterol inclusion complexes could not be delivered at higher concentrations due to deleterious effects on the adhesion and viability of neural progenitors. Taken together, these observations suggest that MβCD:cholesterol is sufficient to activate low-level Hh signals in neural progenitors and consequently to direct differentiation towards neural cell types that depend on such signals.

## Discussion

To establish a causal or regulatory role for a component in a biological pathway, experiments should demonstrate that the component is both *necessary* and *sufficient* for activity. Cholesterol has been shown to be necessary for SMO activation, based on experiments using inhibitors of cholesterol biosynthesis and high concentrations of naked MβCD to strip the plasma membrane of cholesterol (*Cooper et al., 2003*). Impaired SMO activation caused by cholesterol deficiency has also been noted in Smith-Lemli-Opitz syndrome (SLOS), a congenital malformation syndrome caused by defects in the enzyme that converts 7-dehydrocholesterol to cholesterol (*Blassberg et al., 2016*; *Cooper et al., 2003*). In contrast to our results, the SMO CRD is dispensable for this permissive role of cholesterol. The depletion of cholesterol reduces signaling by SMO mutants lacking the entire CRD (*Myers et al., 2013*) or carrying mutations in the CRD binding-groove (*Blassberg et al., 2016*). By analogy with other GPCRs, these permissive effects are likely to be mediated by the SMO 7TMD.

We now find that cholesterol is also sufficient to activate Hh signalling in a dose-dependent manner. This instructive effect is mediated by the Class F GPCR SMO and maps to its extracellular CRD. Cholesterol engages a hydrophobic groove on the surface of the CRD, a groove that was previously shown to mediate the activating influence of oxysterols (*Myers et al., 2013*; *Nachtergaele et al., 2013*; *Nedelcu et al., 2013*) and represents an evolutionarily conserved mechanism for detecting hydrophobic small-molecule ligands (*Bazan and de Sauvage, 2009*). An analogous mechanism is present in the Frizzled family of Wnt receptors, where the Frizzled CRD binds to the palmitoleyl group of Wnt ligands, an interaction that is required for Wnt signaling (*Janda et al., 2012*). Thus, the instructive effects of cholesterol revealed in our present study and the permissive effects of cholesterol reported previously map to distinct, separable SMO domains. Our observation that MβCD:cholesterol is sufficient to activate SMO through its CRD is in agreement with a report published recently during the review process of our manuscript (*Huang et al., 2016*).

There are many reasons why this activating effect of cholesterol on Hh signalling may not have been appreciated previously despite the fact that the activating effects of side-chain oxysterols have

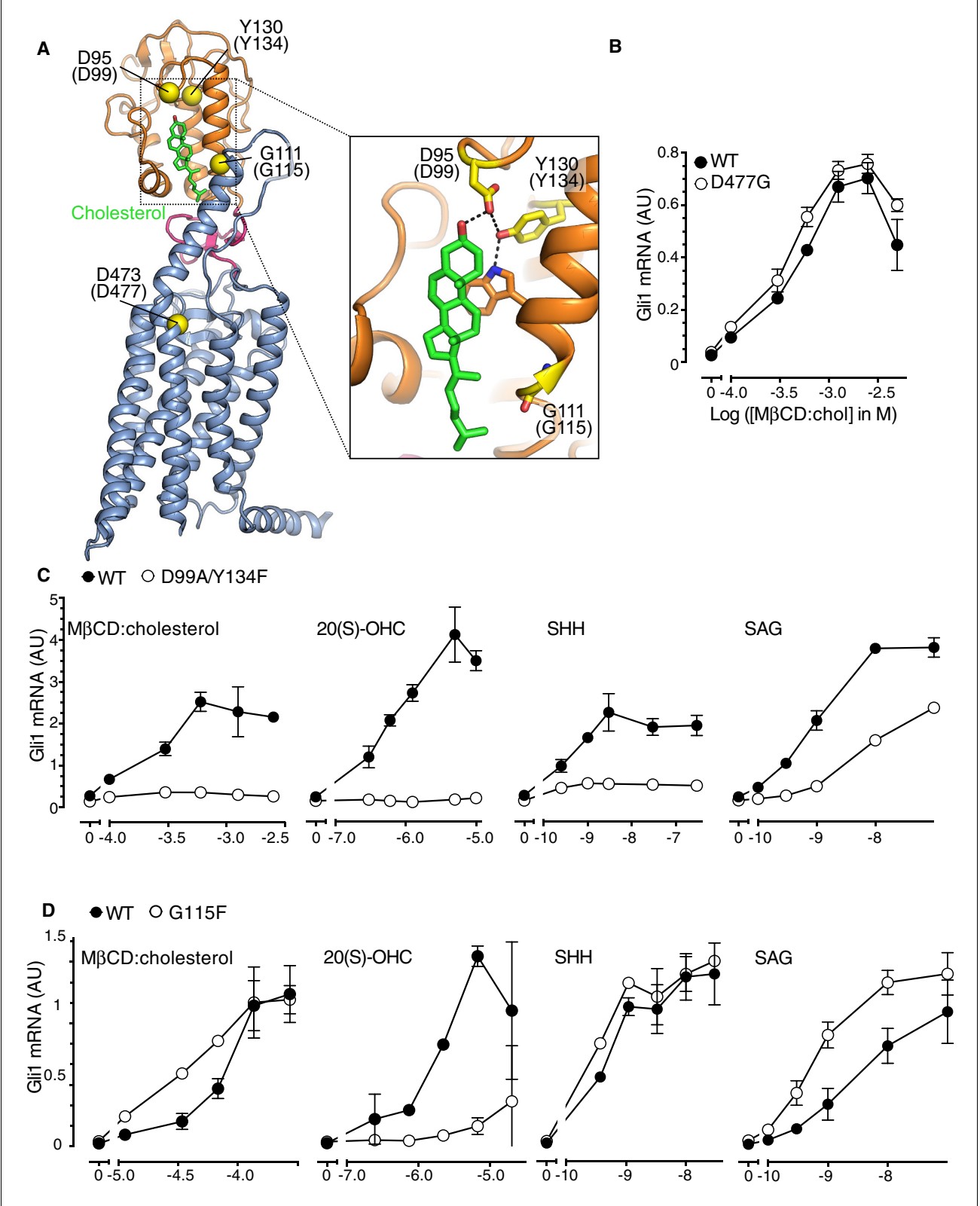

**Figure 4.** The Smoothened cysteine-rich domain is required for cholesterol-mediated activation of Hh signaling. (**A**) Structure of human SMO (PDB 5L7D), with the CRD in orange, the 7TMD in blue, the linker domain in pink, and the cholesterol ligand bound to the CRD in green. The Cα positions of the gatekeeper residues in the two ligand binding sites are highlighted as yellow spheres and numbered, with the mouse numbering shown in parenthesis. The inset shows a close-up of the cholesterol-binding site. D95 and Y130 form part of a hydrogen-bonding network (dotted lines) with the

*Figure 4 continued on next page*

Figure 4 continued

3-hydroxyl of cholesterol, G111 abuts the iso-octyl chain of cholesterol, and D473 is a critical residue in the 7TMD binding-site. (B, C and D) Dose-response curves for the indicated agonists in Smo$^{-/-}$ cells stably expressing WT SMO (always solid black circles) or the indicated SMO variants (open circles) carrying mutations in the 7TMD ligand-binding site (B) or at two opposite ends of the CRD binding groove (C and D). All agonists were applied to cells for 12 hr and mean (±SD) values for *Gli1* mRNA are plotted based on 3 replicates. In C and D, values on the abscissa represent Log ([Agonist] in M) and the ordinate for all four graphs is only shown once at the left.

The following figure supplement is available for figure 4:

**Figure supplement 1.** Role of the cysteine-rich domain of Smoothened in responses to cholesterol and side-chain oxysterols.

been known for a decade (*Corcoran and Scott, 2006*; *Dwyer et al., 2007*). First, the method of delivery, as an inclusion complex with MβCD, is critical to presenting cholesterol, a profoundly hydrophobic and insoluble lipid, in a bioavailable form capable of activating Smo. Even clear solutions of cholesterol in the absence of carriers like MβCD contain microcrystalline deposits or stable micelles that sequester cholesterol (*Haberland and Reynolds, 1973*). In contrast, side-chain oxysterols, which harbor an additional hydroxyl group, are significantly more hydrophilic and soluble in aqueous solutions, shown by their ~ 50 fold faster transfer rates between membranes (*Theunissen et al., 1986*). Second, cholesterol levels in the cell are difficult to manipulate because they are tightly controlled by elaborate homeostatic signalling mechanisms (*Brown and Goldstein, 2009*). MβCD:cholesterol inclusion complexes have been shown to be unique in their ability to increase the cholesterol content of the plasma membrane rapidly at timescales (~1–4 hr) at which cytoplasmic signaling pathways operate (*Christian et al., 1997*; *Yancey et al., 1996*). Other methods of delivery using low density lipoprotein particles and lipid dispersions, or mutations in genes regulating cholesterol homeostasis, function on a much slower time scale and are thus more likely to be confounded by indirect effects given the myriad cellular processes affected by cholesterol (*Christian et al., 1997*). Finally, the bell-shaped Hh signal-response curve (*Figure 1A*) implies that MβCD:cholesterol must be delivered in a relatively narrow, intermediate concentration range (1–2 mM) to observe optimal activity, with higher (>5 mM) concentrations commonly used to load cells with cholesterol producing markedly lower levels of signaling activity.

Our results are particularly informative in light of the recently solved crystal structure of a SMO protein containing the CRD, linker domain and entire 7TMD but lacking the cytoplasmic tail (hereafter called SMO△C) (*Figure 4A*) (*Byrne et al., 2016*). SMO△C was unexpectedly found to contain a cholesterol ligand in its CRD groove. Cholesterol also made key contacts with the linker domain and third extracellular loop of the 7TMD (*Figure 4A*), and molecular dynamics simulations showed that cholesterol can stabilize these extracellular regions of SMO (*Byrne et al., 2016*). However, the function of this bound cholesterol, whether it is an agonist, antagonist or co-factor, remains an important unresolved question in SMO regulation. Structure-guided point mutations in CRD residues that form hydrogen-bonding interactions with the 3β-hydroxyl of cholesterol, reduced signaling by cholesterol (*Figure 4C*), making it likely that cholesterol activates SMO by binding to the CRD in the pose revealed in the structure (*Figure 4A*). Thus, the cholesterol-bound SMO structure recently reported by our groups may very well represent an active-state conformation of the CRD. A comparison of this cholesterol-bound structure with a structure of inactive SMO bound to the potent 7TMD antagonist vismodegib (which lacks cholesterol in the CRD groove) revealed a conformational change that may drive SMO activation (*Figure 6*) (*Byrne et al., 2016*). Cholesterol binding is predicted to induce a clockwise rotation of the CRD on the 7TMD pedestal, perhaps driving SMO activation by a rearrangement of contacts between the CRD and the 7TMD. A caveat to this model is that it depends on a structural comparison with SMO△C bound to a synthetic antagonist and not with un-liganded SMO△C, which has thus far eluded crystallization.

Based on structures of the isolated *Xenopus laevis* CRD, either alone (*apo*-CRD) or in complex with 20(S)-OHC (but notably not cholesterol), a recently published report proposed that sterols drive SMO activation by inducing a conformational change within the CRD itself (*Huang et al., 2016*). We disagree with this model for several reasons. First, these structures do not contain the linker domain and the entire 7TMD, both of which make critical contacts with cholesterol and the CRD, and hence cannot reveal changes in orientation between the CRD and the 7TMD that are essential to

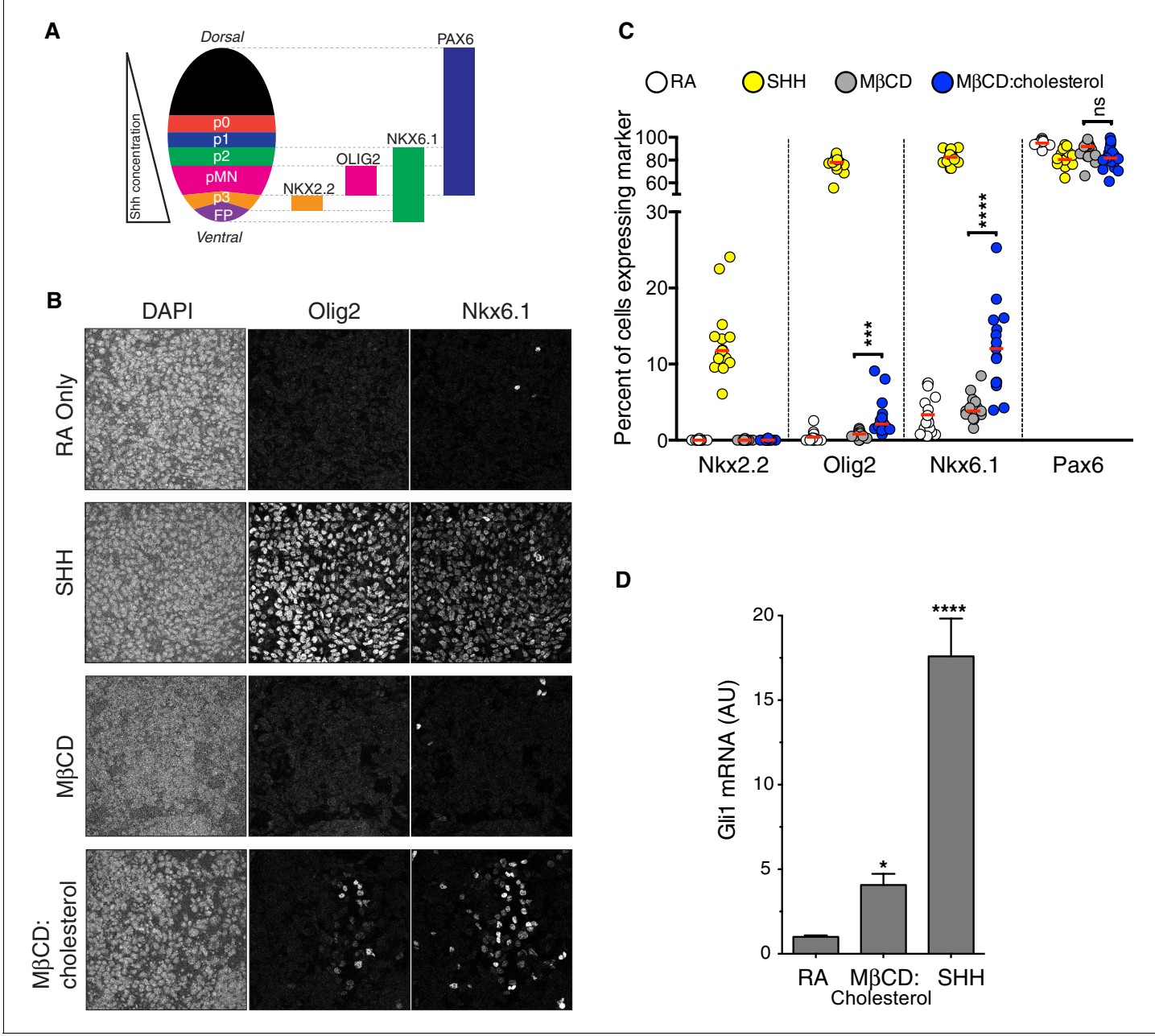

**Figure 5.** Cholesterol induces the differentiation of neural progenitors. (**A**) A schematic illustrating the relationship between marker proteins used to assess differentiation and progenitor cell populations in the embryonic neural tube (taken from (**Niewiadomski et al., 2014**)). FP – floor plate progenitors, MN – motor neuron progenitors, p0, p1, p2, p3 – ventral interneuron progenitors. (**B**) Differentiation of neural progenitors was assessed by immunostaining for Nkx6.1+ and Olig2+ expression (see **A**) after treatment (48 hr) with Retinoic Acid (RA, 100 nM) alone or RA plus SHH (25 nM), MβCD (2 mM) or the saturated MβCD:cholesterol inclusion complex (2 mM). The percentage of nuclei (stained with DAPI) positive for four differentiation markers (see **A**) in 15 different images is plotted in (**C**), with each point representing one image of the type shown in (**B**) and the red line drawn at the median value. Asterisks denote statistical significance (unpaired *t*-test, Holm-Sidak correction, n = 15) for the comparison between cells treated with RA +MβCD and RA+MβCD:cholesterol. (**D**) *Gli1* mRNA (mean ± SD, n = 3) after 48 hr of the indicated treatments. Asterisks denote statistical significance for difference from the RA-treated sample using one-way ANOVA with a Holm-Sidak post-test.

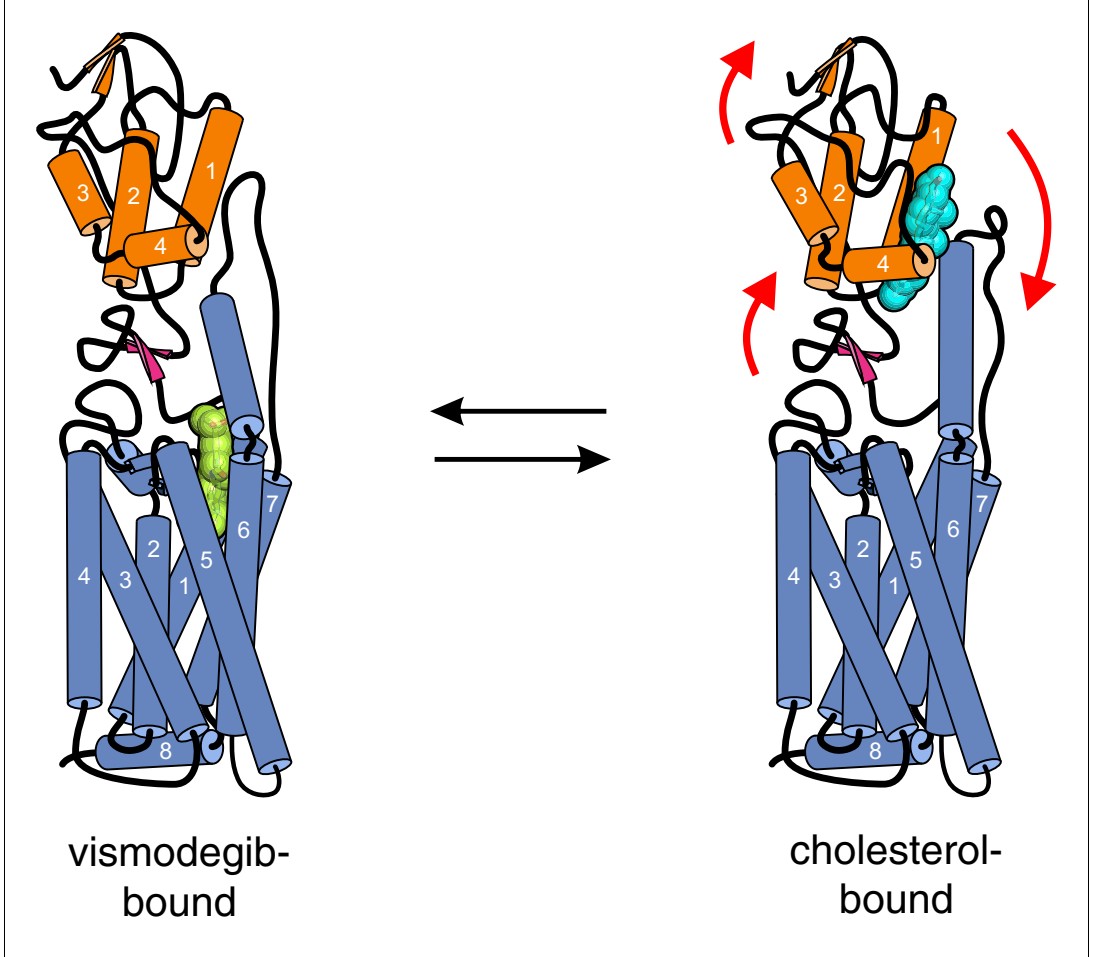

vismodegib-
bound

cholesterol-
bound

**Figure 6.** Conformational changes in Smoothened induced by cholesterol binding. A comparison of the structures of SMO△C bound to vismodegib (PDB ID 5L7I, left), representing an inactive state, and cholesterol (PDB ID 5L7D, right), highlights a rotation of the CRD and the helical extracellular loop 3 relative to the 7TMD. This rotation may communicate ligand binding at the CRD to conformational changes in the 7TMD.

The following figure supplement is available for figure 6:

**Figure supplement 1.** A comparison of all Smoothened CRD structures.

understanding how CRD ligands communicate with the 7TMD. Second, all structures of the SMO CRD (with the exception of the *Xenopus apo*-CRD) are conformationally identical (*Figure 6—figure supplement 1A*), regardless of whether they contain a bound ligand (cholesterol-bound SMO△C, cyclopamine-bound CRD or 20(S)-OHC-bound CRD) or not (*Danio rerio apo*-CRD or vismodegib-bound SMO△C) and regardless of whether they were crystallized by standard vapor diffusion (the CRD structures) or lipidic cubic phase methods (the SMO△C structures). Hence this conserved conformation, seen in both the isolated CRD and the more physiological SMO△C molecule, is unlikely to be an artefact of crystal packing as suggested by these authors (*Huang et al., 2016*). Finally, a careful inspection of the crystal lattice contacts in the *Xenopus apo*-SMO structure (PDB ID 5KZZ) revealed that the region of the proposed conformational change is partially disordered (*Figure 6—figure supplement 1A*) and involved in the coordination of a zinc ion together with a symmetry-related molecule in the crystal, a very tight, near-covalent interaction that was likely driving crystal formation (*Figure 6—figure supplement 1B*). Since the crystallization solution for the *Xenopus apo*-SMO, but not the solutions used to crystallize the sterol-bound CRDs, contained 200 mM zinc acetate, the altered conformation observed may have been induced by a non-physiological, zinc-promoted crystal contact (*Huang et al., 2016*).

A surprising feature of the structure is that CRD-bound cholesterol is located at a considerable distance (~12 Å) away from the membrane, which would require a cholesterol molecule to desolvate from the membrane and become exposed to water in order to access its CRD binding pocket (*Byrne et al., 2016*) (*Figure 7*). The kinetic barrier, or the activation energy ($\Delta G^{\ddagger}$) for this transfer reaction is predicted to be high (~20 kcal/mole), based on the $\Delta G^{\ddagger}$ for cholesterol transfer between two acceptors through an aqueous environment (*Yancey et al., 1996*). The unique ability of MβCD to shield cholesterol from water while allowing its rapid transfer to acceptors would allow it to bypass this kinetically unfavorable step by delivering it to the CRD binding site (*Figure 7*). These considerations present a regulatory puzzle for future research: how does cholesterol gain access to the CRD-binding pocket without MβCD and is this process regulated by native Hh ligands? Indeed, the kinetic barrier for cholesterol transfer to the CRD pocket makes it an ideal candidate for a rate-limiting, regulated step controlling SMO activity in cells.

MβCD:cholesterol was consistently less active than the native ligand SHH in our assays (*Figures 1D*, *5C and D*). Comparing the doses of MβCD:cholesterol to the doses of SHH delivered to cells is difficult. SHH was used at saturating concentrations; however, we could not assess the effects of MβCD:cholesterol at saturating doses, because the downward phase of the bell-shaped dose-response curve (in cultured fibroblasts, *Figure 1A*) and cell toxicity (in neural progenitors) proved to be dose-limiting. Aside from these technical considerations related to delivery, other possibilities for lower activity include the observation that MβCD:cholesterol did not induce the high-level accumulation of SMO in primary cilia (*Figure 3—figure supplement 1*) and the possibility that a different ligand regulates high-level signaling by SMO. Mutations in the 7TMD binding-site do not alter the constitutive or SHH-induced signaling activity of SMO, which has led to view that this site does not regulate physiological signaling (*Myers et al., 2013*; *Yauch et al., 2009*). In contrast, mutations in the cholesterol-binding site impaired responses to SHH (*Byrne et al., 2016*). Hence, a putative alternate ligand would have to engage a third, undefined site. Lastly, the presence of active PTCH1 is a major difference between SHH- and MβCD:cholesterol-induced signaling. The biochemical activity of PTCH1 (which is inactivated by SHH) may oppose the effects of MβCD:cholesterol, limiting signaling responses. Interestingly, MβCD:cholesterol was able to restore maximal Hh responses in the absence of PTCH1 (*Figure 3D*).

Our results may have implications for understanding how PTCH1 inhibits SMO, a longstanding mystery in Hh signaling. The necessity and sufficiency of cholesterol for SMO activation, mediated through two different regions of the molecule, means that SMO activity is likely to be highly sensitive to both the abundance and the accessibility of cholesterol in its membrane environment. Furthermore, PTCH1 has homology to a lysosomal cholesterol transporter, the Niemann-Pick C1 (NPC1) protein (*Carstea et al., 1997*), and PTCH1 has been purported to have cholesterol binding and transport activity (*Bidet et al., 2011*). Thus, our work supports a model where PTCH1 may inhibit SMO by reducing cholesterol content or cholesterol accessibility (or chemical activity) in a membrane compartment that also contains SMO, leading to alterations in SMO conformation or trafficking (*Bidet et al., 2011*; *Incardona et al., 2002*; *Khaliullina et al., 2009*). Since cholesterol is a ubiquitous component of cellular membranes that affects many cellular processes, PTCH1-induced changes in cholesterol are likely to be confined to a specific membrane compartment. The base of the cilium is a good candidate for such a compartment because PTCH1 is localized most prominently at the ciliary base in Hh-responsive tissues in the mouse embryo (*Rohatgi et al., 2007*) and because most Hh pathway components, from PTCH1 to the GLI transcription factors, are found in and around primary cilia (*Briscoe and Thérond, 2013*). Two key questions must be answered before endogenous cellular cholesterol can be considered the elusive second messenger that communicates the Hh signal from PTCH1 to SMO: Do Hh ligands alter cholesterol abundance or activity? and Is cholesterol a substrate for the predicted transporter activity of PTCH1? Answering these questions will require developing or adapting tools to measure and perturb cholesterol in specific cellular compartments and an assessment of the biochemical activities of purified SMO and PTCH1 reconstituted into cholesterol-containing membranes.

While cholesterol is an abundant lipid, clearly critical for maintaining membrane biophysical properties and for stabilizing membrane proteins, our work suggests that it may be also used as a second messenger to instruct signaling events at the cell surface through GPCRs and perhaps other cell-surface receptors.

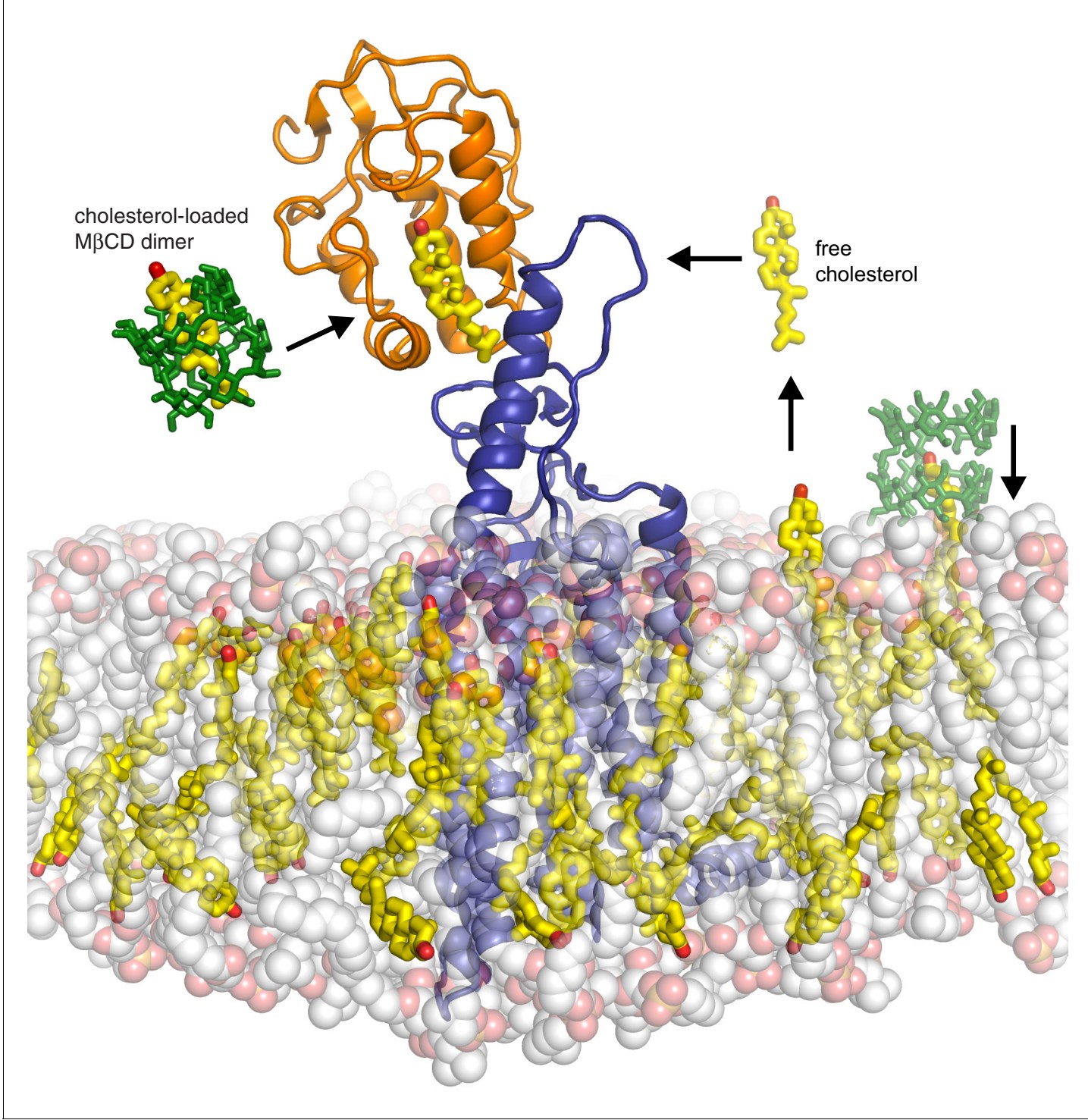

**Figure 7.** Models for how cholesterol may gain access to its binding-site in the SMO cysteine-rich domain. The structure of SMO bound to cholesterol (PDB 5L7D) is shown embedded in a lipid bilayer composed of 1-palmitoyl-2-oleoyl-sn-glycero-3-phosphocholine (POPC) and cholesterol in a ratio of 3:1 (**Byrne et al., 2016**). The SMO CRD is colored orange; the linker domain and 7TMD are colored blue. Two molecules of MβCD (PDB QKH, shown as green sticks) form an inclusion complex with each molecule of cholesterol (PDB CLR, colored yellow in stick representation with the 3-hydroxyl shown red). MβCD could deliver cholesterol directly to the CRD binding pocket (left) or to the outer leaflet of the plasma membrane (right), which would subsequently require a second transfer step from the membrane to the CRD. The activation energy for the direct delivery mechanism on the left (<10 kcal/mole) is much lower than for the mechanism on the right (~20 kcal/mole), where cholesterol has to desolvate from the membrane without a carrier to access the CRD site (**Lopez et al., 2011**; **Yancey et al., 1996**).

## Materials and methods

### Cells and reagents

#### Reagents and cell lines

NIH/3T3 and 293 T cells were obtained from ATCC (Manassas, Virginia), $Smo^{-/-}$ fibroblasts have been described previously (*Varjosalo et al., 2006*) and were originally obtained from Drs. James Chen and Philip Beachy. All cell lines were confirmed to be negative for mycoplasma contamination by PCR. NIH/3T3 and 293 T cells were used within 5 passages of receipt from ATCC and were not re-authenticated by STR profiling in our laboratory. 293 T cells were solely used for viral production and not for the collection of Hh signalling data presented in the manuscript. NIH/3T3 cells were tested by immunoblotting to confirm expression of Hh pathway components (SMO, PTCH1, SUFU, GLI1) and by *Gli*1 qRT-PCR to confirm Hh-pathway responsiveness. $Smo^{-/-}$ fibroblasts were confirmed to lack SMO protein expression by immunoblotting. Suppliers for chemicals included Enzo Life Sciences (SAG; Farmingdale, NY), Toronto Research Chemicals (Cyclopamine; Canada), from Millipore (SANT-1; Hayward, CA), Tocris (20(S)-OHC; United Kingdom), LC Labs (Vismodegib; Woburn, MA), Steraloids (25-OHC, 27-OHC, *epi*-cholesterol; Newport, RI), Sigma-Aldrich (cholesterol, desmosterol, lathosterol, 7-dehydrocholesterol, Methyl-$\tilde{\beta}$cyclodextrin; St. Louis, MO), and Thermo Fisher Scientific (Alexa Fluor 647 NHS ester; Waltham, MA). *Ent*-cholesterol was synthesized as described previously (*Jiang and Covey, 2002*). Antibodies against GLI3 and GLI1 were from R&D Systems (AF3690; Minneapolis, MN) and Cell Signaling Technologies (Cat#L42B10; Danvers, MA) respectively. Human SHH carrying two isoleucine residues at the N-terminus and a hexahistidine tag at the C-terminus was expressed in *Escherichia coli* Rosetta(DE3)pLysS cells and purified by immobilized metal-affinity chromatography followed by gel-filtration chromatography as described previously (*Bishop et al., 2009*). Perfringolysin O (PFO) was purified as previously described (*Das et al., 2013*; *Li et al., 2015*) and covalently labeled with Alexa Fluor 647 dye following the manufacturer's instructions (Thermo Fisher Scientific).

#### Methyl-β-Cyclodextrin sterol complexes

Sterols were dissolved in a mixture of chloroform-methanol (2:1 vol/vol) to generate a 10 mg / mL stock solution. To a glass vial, 8.7 μmole of sterol was delivered from the organic stock solution. Nitrogen gas was streamed over the sterol solution until the organic solvent was evaporated completely, generating a thin film in the vial. MβCD was dissolved in Opti-MEM at a final concentration of 50 mg / mL (38 mM), and 2 mL of this solution was added to the dried sterol film in the glass vial. A micro-tip sonicator was used to dissolve the mixture until it became clear. Solutions were filtered through a 0.1 μm filter and stored in glass vials at 4°C. Unless otherwise stated, the MβCD: cholesterol ratio was 8.8:1 in inclusion complexes. Preparation of the different ratios of cholesterol to MβCD (*Figure 2*) was achieved following the aforementioned protocol changing only the initial molar amount of cholesterol keeping the molar concentration of MβCD constant.

#### Constructs

Constructs encoding mutant mouse SMO (D99A/Y134F, G115F, V333F, D477G, D477R, D477R/Y134F) were generated using the QuikChange method in the pCS2+:mSmo vector (*Byrne et al., 2016*) and then transferred by Gibson cloning to a retroviral vector (pMSCVpuro) for stable cell line construction.

#### Stable cell lines

Stable cell lines were prepared as described previously by infecting $Smo^{-/-}$ mouse embryonic fibroblasts with a retrovirus carrying untagged Smo variants cloned into pMSCVpuro (*Byrne et al., 2016*; *Rohatgi et al., 2009*). Retroviral supernatants were produced after transient transfection of Bosc23 helper cells with the pMSCV constructs (*Pear et al., 1993*). Virus-containing media from these transfections was directly used to infect $Smo^{-/-}$ fibroblasts, and stable integrants were selected with puromycin (2 μg/mL). Cell lines stably expressing SMO-D99A/Y134F, SMO-V333F, SMO-D477R, and SMO-△CRD have been described and characterized previously, including measurement of SMO protein levels by immunoblotting (*Byrne et al., 2016*).

## Hedgehog signaling assays using quantitative RT-PCR

Stable cell lines expressing SMO variants or NIH/3T3 cells were grown to confluency in Dulbecco's Modified Eagle's Medium (DMEM) containing 10% Fetal Bovine Serum (FBS, Optima Grade, Atlanta Biologicals; Flowery Branch, GA). Confluent cells were exchanged into 0.5% FBS DMEM for 24 hr to allow ciliogenesis prior to treatment with drugs and/or ligands in DMEM containing 0.5% FBS for various times, as indicated in the figure legends. The mRNA levels of *Gli1*, a direct Hh target gene commonly used as a metric for signalling strength, were measured using the *Power SYBR Green Cells-To-CT* kit (Thermo Fisher Scientific). The primers used are *Gli1* (forward primer: 5'-ccaagc-caactttatgtcaggg-3' and reverse primer: 5'-agcccgcttctttgttaatttga-3'), *Gapdh* (forward primer: 5'-agtggcaaagtggagatt-3' and reverse primer: 5'-gtggagtcatactggaaca-3'), *Hmgcr* (forward primer: 5'-tgtggtttgtgaagccgtcat-3' and reverse primer: 5'-tcaaccatagcttccgtagttgtc-3'), and *Hmgcs1* (forward primer: 5'-gggccaaacgctcctctaat-3' and reverse primer: 5'-agtcataggcatgctgcatgtg-3'). Transcript levels relative to Gapdh were calculated using the $\triangle$Ct method. Each qRT-PCR experiment, which was repeated 3–4 times, included two biological replicates, each with two technical replicates.

## Data analysis

Each experiment shown in the paper was repeated at least three independent times with similar results. All data was analyzed using GraphPad Prism. All points reflect mean values calculated from at least 3 replicates and error bars denote standard deviation (SD). The statistical tests used to evaluate significance are noted in the figure legends. Statistical significance in the figures is denoted as follows: ns: $p > 0.05$, $*p \leq 0.05$, $**p \leq 0.01$, $***p \leq 0.001$, $****p \leq 0.0001$.

## Mouse embryonic stem cell culture and cell differentiation

For maintenance, MM13 mouse embryonic stem cells (mESCs) (*Wichterle et al., 2002*) were plated on irradiated primary mouse embryonic fibroblasts (pMEFs) and cultured in mESC media (Dulbecco's Modified Eagle's Medium high glucose (Hyclone; Pittsburgh, PA) and 15% Optima FBS (Atlanta Biologicals) supplemented with 1% MEM non-essential amino acids (Thermo Fisher Scientific), 1% penicillin/streptomycin (Gemini Bio-Products; West Sacramento, CA), 2 mM L-glutamine (Gemini Bio-Products), 1% EmbryoMax nucleosides (Millipore), 55 µM 2-mercaptoethanol (Thermo Fisher Scientific), and 1000 U/ml ESGRO LIF (Millipore). The mESCs were differentiated as previously described with minor modifications (*Gouti et al., 2014*; *Ying et al., 2003*). Briefly, the pMEFs were removed from the mESCs by dissociating the cells with 0.25% Trypsin/EDTA and then incubating the cells on tissue culture plates for two short successive periods (20 min each). To induce differentiation, the cells were plated on Matrigel (BD Biosciences; San Jose, CA) coated glass coverslips (12 mm diameter, placed in a 24-well plate) at a density of $2.4 \times 10^4$ cells per coverslip in N2B27 media (Dulbecco's Modified Eagle's Medium F12 (Gibco) and Neurobasal Medium (Gibco) (1:1 ratio) supplemented with N-2 Supplement, B-27 Supplement, 1% penicillin/streptomycin), 2 mM L-glutamine , 40 µg/ml Bovine Serum Albumin, and 55 µM 2-mercaptoethanol). On Day 0 and Day 1, cells were cultured in N2B27 with 10 ng/ml bFGF (R&D Scientific). On Day 2, the media was changed and the cells were cultured in N2B27 with 10 ng/ml bFGF (R&D Scientific) and 5 µM CHIR99021 (Axon; Netherlands). On Day 3, the media was changed and the cells were cultured in 1 ml of N2B27 supplemented with 100 nM Retinoic Acid (RA), 100 nM RA and 25 nM SHH, 100 nM RA and 2 mM MeβCD, or 100 nM RA and 2 mM MβCD + 0.23 mM cholesterol. On Day 4, 1 ml N2B27 with 100 nM RA was added to each well, thus diluting each treatment condition by half. On Day 5 the cells were rinsed and fixed for further analysis.

## Immunofluorescence

NIH/3T3 cells were cultured in Dulbecco's modified Eagle's Medium (DMEM) containing 10% Fetal Bovine Serum (FBS, Optima Grade, Atlanta Biologicals) in 24-well plates at an initial density of $7.5 \times 10^4$ on acid-washed glass cover-slips that were pre-coated with poly-L-lysine. Confluent cells were exchanged into 0.5% FBS DMEM to induce ciliogenesis for 24 hr. Ciliated cells were treated with the indicated drugs each dissolved in 0.5% FBS DMEM. Samples were fixed using 4% paraformaldehyde in phosphate buffered saline (PBS) for 10 min and washed three times with PBS. For SMO localization studies, cells were blocked and permeabilized in 1% donkey serum, 10 mg / mL bovine serum albumin (BSA), 0.1% triton X-100, and PBS. Primary antibodies were administered in

block buffer for 2 hr at room temperature. Cover-slips were washed three times with wash buffer containing PBS and 0.1% triton X-100. Secondary antibodies were administered in block buffer for 1 hr. Cover-slips were washed three more times in wash buffer and mounted on glass slides using Pro-Long Diamond Antifade Mountant with DAPI (Thermo Fisher Scientific). For PFO staining, cells were fixed in 4% PFA, washed three times with PBS and stained with PFO in PBS without detergent. Cover-slips were washed three times with PBS and mounted on glass slides using Pro-Long Diamond Antifade Mountant with DAPI (Thermo Fisher Scientific). Images were acquired on an inverted Leica SP8 laser scanning confocal microscope with a 63X oil immersion objective (NA 1.4) using a HyD hybrid detector. Z-stacks were acquired with identical acquisition settings (gain, offset, laser power, frame format) within a given experiment. The following primary antibodies were used: rabbit anti-Smo (1:500) (*Rohatgi et al., 2007*), guinea pig anti-Arl13b (*Pusapati et al., 2014*), goat anti-GFP (1:2000) (Rockland; Limerick, PA), mouse anti-Nkx2.2 (1:100) (74.5A5, Developmental Studies Hybridoma Bank, Iowa City, IA), mouse anti-Nkx6.1 (1:100) (F55A10, Developmental Studies Hybridoma Bank), guinea pig anti-Olig2 (1:20,000) (*Novitch et al., 2001*), rabbit anti-Pax6 (1:1000) (AB2237, Millipore). The following secondary antibodies were used: Alexa Fluor 488, Alexa Fluor 594, and Alexa Fluor 647 (Thermo Fisher Scientific).

## Image analysis

Image processing for ciliary SMO levels was carried out using maximum projection images of the acquired Z-stacks using ImageJ. For quantification of ciliary Smo, first a mask was constructed using the Arl13b image (primary cilia marker), and then the mask was applied to the corresponding Smo image where the integrated fluorescence was measured. An identical region outside the cilia was measured to determine background fluorescence. Background correction was applied on a per cilia basis by subtracting the background fluorescence from the cilia fluorescence.

For neural differentiation experiments, fluorescent images were collected on a Leica TCS SP8 confocal imaging system equipped with a 40x oil immersion objective using the Leica Application Suite X (LASX) software. For each experiment, coverslips from each condition were grown, collected, and processed together to ensure that the cells were fixed and stained for the same duration of time. To ensure uniformity in imaging, the gain, offset, and laser power settings on the microscope were held constant for each antibody. At least 15 images were taken per condition. To ensure all cells were represented, z-stacks were acquired and counts were performed on the compressed images. Cell counts were conducted using the NIH ImageJ software suite with cell counter plugin. In total, 5000–6000 cells were analyzed per condition. The experiment was conducted independently a total of three times. Representative images shown in *Figure 5* were processed equally using Adobe Photoshop, Adobe illustrator, and CorelDraw software.

## Cholesterol quantification

Cells were cultured in Dulbecco's modified Eagle's Medium (DMEM) containing 10% Fetal Bovine Serum (FBS, Optima Grade, Atlanta Biologicals) in 6-well plates at an initial density of $3 \times 10^5$ cells / well. Confluent cells were switched into 0.5% FBS DMEM to induce ciliogenesis for 24 hr. Cells were treated with indicated drugs dissolved in 0.5% FBS DMEM in duplicate. One sample was used to measure total protein by bicinchoninic acid assay (BCA), and the second for total lipid extraction and subsequent cholesterol quantification. Cells were washed once with Phosphate Buffered Saline (PBS), and harvested using a Corning cell lifter in PBS. The cell suspension was transferred to a 1.5 mL ependorf tube, centrifuged at 1000 x g and the PBS aspirated. Total lipids were extracted from the cell pellet by the addition of 200 μL of chloroform-methanol (2:1 vol/vol). To induce phase separation, 100 μL of PBS was added to the lipid extract and the sample was centrifuged at 5000 x g for 5 min. The organic layer was transferred to a fresh 1.5 mL eppendorf tube and the solvent removed under reduced pressure. Relative total free cholesterol was measured using the *Amplex Red Cholesterol Assay Kit* (Thermo Fisher Scientific) following the manufacturer's instructions. Lysis buffer containing 50 mM Tris pH 7.4, 150 mM NaCl, 2% Nonidet P-40, 0.5% sodium deoxycholate, 0.1% sodium dodecyl sulfate, 1 mM dithithreitol, and Sigma Fast protease inhibitor cocktail (Sigma-Aldrich) was used to disrupt the cell pellet. A ratio of total free cholesterol to total protein was used as a normalization method.

## Acknowledgements

We thank Andres Lebensohn, Ganesh Pusapati, James Briscoe, Robert Blassberg and George Hedger for discussions and comments on the manuscript. This work was supported by the US National Institutes of Health (GM106078 and HL067773), Cancer Research UK (C20724/A14414), and the Taylor Family Institute for Psychiatric Research. GL was supported by the Ford Foundation, SN by the National Science Foundation and EFXB by NDM Oxford. AS has received funding from an EMBO LTF (1438–2013), HFSP LTF (LT000401/2014 L) and the People Programme (Marie Curie Actions) of the European Union's Seventh Framework Programme FP7-2013 under REA grant agreement n° 624973.

## Additional information

### Funding

| Funder | Grant reference number | Author |
|---|---|---|
| Ford Foundation | | Giovanni Luchetti |
| National Science Foundation | | Sigrid Nachtergaele |
| European Molecular Biology Organization | LTF - 1438-2013 | Andreas Sagner |
| Human Frontier Science Program | LTF - LT00040½014-L | Andreas Sagner |
| European Commission | People Programme of the EU's Seventh Framework Programme - FP7-2013 | Andreas Sagner |
| Nuffield Department of Medicine, Oxford University | | Eamon FX Byrne |
| National Institutes of Health | GM106078 | Douglas F Covey Rajat Rohatgi |
| Taylor Family institute for Psychiatric Research | | Douglas F Covey |
| National Institutes of Health | HL067773 | Douglas F Covey Rajat Rohatgi |
| Cancer Research UK | C20724/A14414 | Christian Siebold |

The funders had no role in study design, data collection and interpretation, or the decision to submit the work for publication.

### Author contributions

GL, Conception and design, Acquisition of data, Analysis and interpretation of data, Drafting or revising the article, Contributed unpublished essential data or reagents; RS, JHK, SN, Acquisition of data, Analysis and interpretation of data, Drafting or revising the article, Contributed unpublished essential data or reagents; AS, EFXB, DFC, Analysis and interpretation of data, Drafting or revising the article, Contributed unpublished essential data or reagents; CS, Conception and design, Analysis and interpretation of data, Drafting or revising the article, Contributed unpublished essential data or reagents; RR, Conception and design, Analysis and interpretation of data, Drafting or revising the article

### Author ORCIDs

Christian Siebold, http://orcid.org/0000-0002-6635-3621
Rajat Rohatgi, http://orcid.org/0000-0001-7609-8858

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
