## [Decision Letter]

Thank you for submitting your article "Cholesterol activates the G-protein coupled receptor Smoothened to promote morphogenetic signaling" for consideration by *eLife*. Your article has been reviewed by three peer reviewers, and the evaluation has been overseen by a Reviewing Editor and Marianne Bronner as the Senior Editor. The reviewers have opted to remain anonymous.

The reviewers have discussed the reviews with one another and the Reviewing Editor has drafted this decision to help you prepare a revised submission.

In this study, Rohatgi and colleagues identify cholesterol as a bona fide activator of Smoothened (Smo) that works through its extracellular cysteine rich domain (CRD) to induce low-level signaling. The hunt for a natural ligand for Smo has been long and largely unsuccessful, making this discovery an impactful advance for the field. Although cholesterol is not likely the sole molecule required for Smo activation (it is insufficient to induce Smo ciliary translocation or high-level signaling), it is an important player in physiological regulation of this GPCR.

Overall, the reviewers find the study to be well-controlled, well-executed and appropriately interpreted. They suggest the following major points for consideration in preparing a revised manuscript.

1) The reviewers noted that a paper from Salic lab with similar conclusions was just published (Huang et al. 2016). Although this detracts from the novelty of this manuscript, the work was clearly performed independently and actually greatly increases the confidence in this important breakthrough. The Salic paper should be cited in the revision. The authors are encouraged to expand the Discussion of the manuscript to highlight differences between the two studies and the important points unique to this manuscript. For example, concerning the mechanism by which cholesterol enhances Smo signaling (i.e. does it induce a CRD conformational shift?), some discussion of the differing structural effects of cholesterol on the CRD of the full-length structure (this group's nature paper) and the soluble CRD (Salic structure) might be informative. That cholesterol does not induce dramatic conformational shifts in the CRD of the full-length protein is now an important issue given the claim of the Salic paper that cholesterol induces a 'dramatic' shift.

2) Figure 3 appears to be n=1. If the conclusions are to be included in the paper, repetitions and statistics should be provided.

---

## [Author Response]

*Overall, the reviewers find the study to be well-controlled, well-executed and appropriately interpreted. They suggest the following major points for consideration in preparing a revised manuscript.*

*1) The reviewers noted that a paper from Salic lab with similar conclusions was just published (Huang et al. 2016). Although this detracts from the novelty of this manuscript, the work was clearly performed independently and actually greatly increases the confidence in this important breakthrough. The Salic paper should be cited in the revision. The authors are encouraged to expand the Discussion of the manuscript to highlight differences between the two studies and the important points unique to this manuscript. For example, concerning the mechanism by which cholesterol enhances Smo signaling (i.e. does it induce a CRD conformational shift?), some discussion of the differing structural effects of cholesterol on the CRD of the full-length structure (this group's Nature paper) and the soluble CRD (Salic structure) might be informative. That cholesterol does not induce dramatic conformational shifts in the CRD of the full-length protein is now an important issue given the claim of the Salic paper that cholesterol induces a 'dramatic' shift.*

The Salic paper (PMID: 27545348), which was not published at the time our manuscript was submitted to *eLife*, is now cited in the revised manuscript. In response to this recommendation, we have expanded the Discussion section to include a comparison of (1) the SMO CRD structures published in the Salic paper, (2) the SMO CRD structure published by us in *eLife* in 2013 (PMID: 24171105) and (3) structures of a complete SMO molecule including the CRD and the entire heptahelical transmembrane bundle (7TMD) published very recently by our groups in *Nature* (PMID: 27437577). It is important to note that the only available structure of cholesterol bound to Smoothened is presented in our *Nature* paper; the Salic study does not show a structure of cholesterol bound to any domain of Smoothened.

Our analysis, depicted in a new Figure 6 and Figure 6—figure supplement 1, leads us to propose that the relevant conformational change upon cholesterol binding involves a rotation of the CRD on the 7TMD pedestal (shown in the new Figure 6). We disagree with the structural conclusions of the Salic paper, derived entirely from structures of the isolated CRD, that SMO activation is driven by a cholesterol-induced conformational change in the CRD itself because (1) this conformational change was not seen in the context of near full-length SMO containing both the CRD and 7TMD (Figure 6—figure supplement 1) and (2) this conformational change seems to be stabilized by a non-physiological, zinc-induced crystal contact (Figure 6—figure supplement 1).

Finally, we have also emphasized some the major limitations of both our studies. Neither our manuscript nor the Salic paper has shown that (1) native Hh ligands modulate cellular cholesterol locally or globally in cells or that (2) Patched 1 regulates cellular/endogenous cholesterol to influence SMO signaling.

*2) Figure 3 appears to be n=1. If the conclusions are to be included in the paper, repetitions and statistics should be provided.*

Figure 3 has been replaced by a revised panel that includes replicates and statistics.